# Identification of Human Enzymes Oxidizing the Anti-Thyroid-Cancer Drug Vandetanib and Explanation of the High Efficiency of Cytochrome P450 3A4 in its Oxidation

**DOI:** 10.3390/ijms20143392

**Published:** 2019-07-10

**Authors:** Radek Indra, Petr Pompach, Václav Martínek, Paulína Takácsová, Katarína Vavrová, Zbyněk Heger, Vojtěch Adam, Tomáš Eckschlager, Kateřina Kopečková, Volker Manfred Arlt, Marie Stiborová

**Affiliations:** 1Department of Biochemistry, Faculty of Science, Charles University, Albertov 2030, CZ-12843 Prague 2, Czech Republic; 2Department of Chemistry and Biochemistry, Mendel University in Brno, Zemedelska 1, CZ-61300 Brno, Czech Republic; 3Department of Pediatric Hematology and Oncology, 2nd Faculty of Medicine, Charles University and University Hospital Motol, V Uvalu 84/1, CZ-150 06 Prague 5, Czech Republic; 4Department of Oncology, 2nd Medical Faculty, Charles University and University Hospital Motol, V Uvalu 84/1, CZ-150 06 Prague 5, Czech Republic; 5Department of Analytical, Environmental and Forensic Sciences, MRC-PHE Centre for Environment and Health, King’s College London, 150 Stamford Street, London SE1 9NH, UK; 6NIHR Health Protection Research Unit in Health Impact of Environmental Hazards at King’s College London in partnership with Public Health England and Imperial College London, 150 Stamford Street, London SE1 9NH, UK

**Keywords:** vandetanib, tyrosine kinase inhibitor, metabolism, cytochromes P450, flavin-containing monoxygenases

## Abstract

The metabolism of vandetanib, a tyrosine kinase inhibitor used for treatment of symptomatic/progressive medullary thyroid cancer, was studied using human hepatic microsomes, recombinant cytochromes P450 (CYPs) and flavin-containing monooxygenases (FMOs). The role of CYPs and FMOs in the microsomal metabolism of vandetanib to *N*-desmethylvandetanib and vandetanib-*N*-oxide was investigated by examining the effects of CYP/FMO inhibitors and by correlating CYP-/FMO-catalytic activities in each microsomal sample with the amounts of *N*-desmethylvandetanib/vandetanib-*N*-oxide formed by these samples. CYP3A4/FMO-activities significantly correlated with the formation of *N*-desmethylvandetanib/ vandetanib-*N*-oxide. Based on these studies, most of the vandetanib metabolism was attributed to *N*-desmethylvandetanib/vandetanib-*N*-oxide to CYP3A4/FMO3. Recombinant CYP3A4 was most efficient to form *N*-desmethylvandetanib, while FMO1/FMO3 generated *N*-oxide. Cytochrome b_5_ stimulated the CYP3A4-catalyzed formation of *N*-desmethylvandetanib, which is of great importance because CYP3A4 is not only most efficient in generating *N*-desmethylvandetanib, but also most significant due to its high expression in human liver. Molecular modeling indicated that binding of more than one molecule of vandetanib into the CYP3A4-active center can be responsible for the high efficiency of CYP3A4 *N*-demethylating vandetanib. Indeed, the CYP3A4-mediated reaction exhibits kinetics of positive cooperativity and this corresponded to the in silico model, where two vandetanib molecules were found in CYP3A4-active center.

## 1. Introduction

Studies carried out over the last decades demonstrate that drugs used for cancer chemotherapy have a narrow therapeutic index, and often the produced responses are only palliative and unpredictable. Although such approaches are directed towards certain biomacromolecules, they often do not discriminate between rapidly dividing tumor vs non-malignant cells [1,2]. In contrast, targeted therapy has been introduced in recent years, which is directed against cancer-specific targets and signaling pathways thus providing more limited nonspecific mechanisms [3]. The most promising targets include receptor tyrosine kinases (TKs), enzymes that selectively phosphorylate hydroxyl moieties of tyrosine residues on signal transduction molecules with a phosphate moiety from adenosine triphosphate [4,5]. Therefore, inhibitors of these enzymes, tyrosine kinase inhibitors (TKIs), are now considered as promising small molecules for the treatment of several types of cancer including the thyroid carcinomas [5].

Vandetanib (Caprelsa, *N*-(4-bromo-2-fluorophenyl)-6-methoxy-7-[(1-methylpiperidin-4-yl)-methoxy]-quinazolin-4-amine; Figure 1) is a TKI—it inhibits vascular endothelial growth factor receptor 2 (VEGFR-2), epidermal growth factor receptor (EGFR) and rearranged during transfection (RET) tyrosine kinase activity [6,7,8,9,10]. It has also activity against protein tyrosine kinase 6 (BRK), tyrosine kinase with immunoglobulin and epidermal growth factor (EGF) domains-2 (TIE2), members of the ephrin receptor kinase family, and members of the SRC family of tyrosine kinases [11]. Therefore, vandetanib is considered a multiple TKI. Its administration reduces tumor cell-induced angiogenesis, tumor vessel permeability, and inhibits tumor growth and metastasis in mouse models of cancer [11]. These activities result from their major targets; VEGFR-2, EGFR and RET, because they are all involved in signaling pathways promoting angiogenesis and tumor growth [10,12,13,14]. Vandetanib is used for the treatment of symptomatic or progressive medullary thyroid cancer, probably because the RET tyrosine kinase mutation occurs in this type of thyroid cancer [11,15,16,17,18]. Treatment with vandetanib is usually well-tolerated by patients but is contraindicated in people with congenital long QT syndrome. In addition, various side effects such as diarrhea, rash and/or nausea can occur [19,20].

In cancer chemotherapy, serious clinical consequences may occur from small alterations in drug metabolism affecting drug pharmacokinetics [21]. Although in general there is only little known about the metabolism of TKIs, the metabolism of vandetanib has been investigated, mainly in studies carried out by AstraZeneca [10,11,20,22,23] but also others [24]. The metabolism of vandetanib in human was shown to be the same as that observed in experimental animal models such as rat, mouse and dog. *N*-desmethylvandetanib and vandetanib-*N*-oxide (Figure 1) were found as major metabolites in excreta of humans and experimental animals [10,20]. In mice and rats, *O*-desmethylvandetanib glucuronide was found as minor metabolite [10,20,24]. This glucuronide conjugate was also detected in human urine. In healthy male subject’s administration of radiolabeled vandetanib confirmed the presence of *N*-desmethylvandetanib and vandetanib-*N*-oxide in plasma, urine and feces [10]. Both metabolites have shown in vitro pharmacologic activity in cellular assays for VEGFR and EGFR [10].

Studies carried out by AstraZeneca have indicated that cytochrome P450 (CYP) and flavin-containing monooxygenase (FMO) enzymes are involved in the metabolism of vandetanib [10,23]. While CYP3A4 was considered as the major enzyme oxidizing vandetanib to *N*-desmethylvandetanib, FMOs expressed in the kidney (FMO1) and liver (FMO3) (Figure 1) were suggested to be responsible for the formation of vandetanib-*N*-oxide [10,23]. However, some other studies did not fully confirm these results, showing contrasting results. For example, itraconazol, a strong inhibitor of CYP3A4 was almost ineffective in inhibiting the formation of *N*-desmethylvandetanib after vandetanib treatment, which would indicate a relative low potency of this enzyme to catalyze this oxidation reaction [10]. Because information on the oxidation of vandetanib by other biotransformation enzymes is absent in the current literature, metabolic studies are needed to further characterize enzymes capable of oxidizing vandetanib. Firstly, these studies are important because vandetanib metabolites exhibit rather different pharmacological efficiencies compared to the parent drug. While *N*-desmethylvandetanib possesses similar potency as vandetanib, vandetanib-*N*-oxide is more than 50-fold less active than the parent drug [10]. Hence, the metabolism of vandetanib might determine its pharmacological (anticancer) efficiency. Secondly, confirming results demonstrating that vandetanib is oxidized by CYP3A4 or identifying additional CYPs (or other biotransformation enzymes) capable of metabolizing vandetanib is important to evaluate whether a patient’s response to vandetanib is influenced by other host factors (e.g., genetic polymorphisms). Inhibition or induction of these biotransformation enzymes by various agents would influence the efficacy of vandetanib during treatment. This is a reasonable scenario because oncology patients frequently take a wide range of other drugs in addition to their therapy for cancer, possibly including compounds that can induce or inhibit enzymes metabolizing anticancer drugs. Significant pharmacokinetic drug–drug interactions can lead to alterations in plasma concentrations of vandetanib, potentially resulting in a reduction in efficacy or an increase in drug-related toxicity.

In the present study we investigated the in vitro metabolism of vandetanib in cell-free systems. We used hepatic subcellular systems (microsomes) isolated from the human livers which are rich in enzymes metabolizing a variety of different drugs and other xenobiotics. In addition, individual recombinant human CYP and FMO enzymes were employed to identify enzymes capable of oxidizing this drug. Because CYP3A4 was identified as the major enzyme metabolizing vandetanib in human liver, the mechanism of the CYP3A4-catalyzed oxidation of vandetanib which determines its pharmacological efficiencies, was characterized in detail. Further, since the heme protein cytochrome b_5_ plays an important role in CYP3A4-mediated microsomal oxidative reactions, for example testosterone 6β-hydroxylation, ellipticine oxidation and nifedipine oxidation [25,26,27,28,29], its effects on vandetanib oxidation by CYP enzymes were also investigated.

## 2. Results

### 2.1. Oxidation of Vandetanib by Human, Rat, Mouse and Rabbit Hepatic Microsomes

First, we investigated the function of human hepatic microsomal subcellular systems containing biotransformation enzymes to catalyze the oxidation of vandetanib and compared it with the efficiency of enzymes present in hepatic microsomes isolated from experimental animals, which were considered as models to mimic the fate of vandetanib in humans [10]. For this comparison human, rat, rabbit and mouse hepatic microsomes were utilized in the present study. All liver microsomes oxidized vandetanib to two metabolites that were separated by HPLC (Figure 2) and identified by mass spectrometry as *N*-desmethylvandetanib and vandetanib-*N*-oxide (Figure 3). The formation of vandetanib metabolites was dependent on nicotinamide adenine dinucleotide phosphate reduced (NADPH) which serves as cofactor for both POR-mediated CYP catalysis and FMO-mediated oxidative reactions in human, rat, rabbit and mouse liver microsomes (Figure 2). Without NADPH no oxidation of vandetanib was detected (Figure 2).

These results demonstrated that the oxidation of vandetanib in hepatic microsomes is mediated by CYPs and/or FMOs. Human hepatic microsomes produced mainly *N*-desmethylvandetanib, while vandetanib-*N*-oxide was formed at ~4-fold lower amounts (*p* < 0.001). In contrast, rat, rabbit and mouse hepatic microsomes oxidized vandetanib to vandetanib-*N*-oxide at 4- to 8-fold higher amounts than *N*-desmethylvandetanib (*p* < 0.001) (Figure 4).

In order to identify individual human CYPs and/or FMOs oxidizing vandetanib and estimate their contribution to the oxidation process, three approaches were utilized: (i) use selective CYP and FMO inhibitors in human hepatic microsomes; (ii) correlation of CYP- and FMO-linked activities in human hepatic microsomes with the amounts of vandetanib metabolites generated by the same microsomes; and (iii) use recombinant human CYPs and FMOs. In addition, analysis of the data on the CYP-mediated formation of *N*-desmethylvandetanib by individual human recombinant CYPs in combination with the CYP enzyme expression levels in human livers was employed to evaluate CYP contributions to this reaction in human livers.

### 2.2. The Effects of CYP and FMO Enzyme Inhibitors on Vandetanib Oxidation in Human Liver Microsomes

The role of human hepatic enzymes in vandetanib oxidation was initially investigated by modulating the enzyme-catalyzed reactions using selective inhibitors of individual CYPs and FMOs in human hepatic microsomes. Two approaches were utilized in the inhibition studies: (i) evaluation of the % inhibition at inhibitor concentrations equimolar to that of the substrate (vandetanib); and (ii) determination of the IC_50_ values for individual inhibitors. Inhibitors α-naphthoflavone (α-NF), diamantane, sulfaphenazole, quinidine, diethyldithiocarbamate (DDTC) and ketoconazole evaluated the CYP1A-, 2B-, 2C-, 2D-, 2E1- and 3A-mediated formation of *N*-desmethylvandetanib, respectively (Table 1). Ketoconazole was the most efficient in inhibiting the oxidation of vandetanib to *N*-desmethylvandetanib, decreasing its formation by 98% at 50 µM. The IC_50_ for ketoconazole equaled to 2 µM (Table 1). Under concentrations equimolar to vandetanib, sulfaphenazole and quinidine inhibited the formation of *N*-desmethylvandetanib by 38% and 20%, respectively (Table 1). The IC_50_ values could not be exactly determined, since even at concentrations of 500 µM, the inhibition was less than 50%. Inhibitors of other CYPs (α-NF, diamantane and DDTC) showed no inhibitory effects on vandetanib oxidation in human microsomes (Table 1).

None of the tested CYP inhibitors was capable of inhibiting the formation of vandetanib *N*-oxide. These results suggest that CYP enzymes do not catalyze the oxidation of vandetanib to its *N*-oxide and that CYP3A is most efficient in catalyzing the oxidation of vandetanib to *N*-desmethylvandetanib.

Methimazole, an inhibitor of FMO [30,31], was used to inhibit FMO-catalyzed oxidation of vandetanib. Formation of vandetanib-*N*-oxide in human liver microsomes was inhibited by 79% at equimolar concentrations to vandetanib, having an IC_50_ value of 6 µM (Table 1). Therefore, FMO can be responsible for oxidation of vandetanib to vandetanib-*N*-oxide in human hepatic microsomes. In this context, it is noteworthy that methimazole also marginally decreased the formation of *N*-desmethylvandetanib in human microsomes (by ~5%), but the observed effect was not statistically significant.

### 2.3. Correlation of CYP- and FMO-Linked Enzyme Activities in Human Hepatic Microsomes with Vandetanib Oxidation to N-Desmethylvandetanib and Vandetanib-N-Oxide

To further identify authentic human CYP and FMO enzymes oxidizing vandetanib, hepatic microsomal samples from livers of 12 different human donors were used in additional experiments. Whereas all of the human microsomal preparations oxidized vandetanib to *N*-desmethylvandetanib, the amounts of vandetanib-*N*-oxide in several samples were below the limit of detection (Table 2). Correlations between the CYP/FMO-catalytic activities and the amounts of *N*-desmethylvandetanib and vandetanib-*N*-oxide formed in each microsomal sample (Table 2) were used to examine the role of specific human CYP and FMO enzymes in their generation. Formation of *N*-desmethylvandetanib highly correlated with testosterone-6β-hydroxylation, a marker for CYP3A4 (*r* = 0.984; *p* < 0.001, Table 3). Further, generation of *N*-desmethylvandetanib also correlated with (S)-mephenytoin-*N*-demethylase, a marker for CYP2B6 (*r* = 0.826; *p* < 0.001), coumarine-7-hydroxylase, a marker for CYP2A6 (*r* = 0.778; *p* < 0.01), and paclitaxel-6α-hydroxylase, a marker for CYP2C8 (*r* = 0.747; *p* <0.01), but not with the activities of other CYPs and FMO. These findings suggest that CYP3A4, followed by CYP2B6, 2A6 and 2C8 enzymes, could be responsible for the formation of *N*-desmethylvandetanib in liver microsomes. However, there is a cross-correlation between testosterone-6β-hydroxylation and (S)-mephenytoin-*N*-demethylase activity (*r* = 0.823; *p* < 0.001), testosterone-6β-hydroxylation and coumarine-7-hydroxylase activity (*r* = 0.786; *p* < 0.001), and testosterone-6β-hydroxylation and paclitaxel-6α-hydroxylase activity (*r* = 0.750; *p* < 0.01), within these liver samples. To additionally clarify these correlations, multivariate analysis was used to investigate the dependence of *N*-desmethylvandetanib formation on these CYP activities. Activities of CYP2B6, 2A6 and 2C8 in each microsomal sample were combined in pairs with the activities of CYP3A4 to see if any combination of the activities led to an improvement in the correlation with vandetanib oxidation to *N*-desmethylvandetanib, i.e., an increase in the correlation coefficient when compared with the correlation with the individual activities. However, the inclusion of the CYP2B6 or 2A6 or 2C8 activities led to no improvement in the correlation coefficient. Therefore, CYP3A4 has the highest impact on the generation of *N*-desmethylvandetanib in human hepatic microsomes.

The formation of vandetanib-*N*-oxide was correlated with methyl-*p*-tolyl sulfide oxidase, a marker for FMO (*r* = 0.736; *p* < 0.01; Table 2), which indicated that vandetanib-*N*-oxide formation is catalyzed by FMO within the human hepatic microsomes. However, we also found a correlation between formation of vandetanib-*N*-oxide and chlorzoxazone-6-hydroxylase, a marker for CYP2E1 (*r* = 0.714; *p* < 0.01) within these liver samples. Although the cross-correlation between methyl-*p*-tolyl sulfide oxidase activity and chlorzoxazone-6-hydroxylase activity (*r* = 0.552) within these liver samples was weak, the observed correlation of the FMO activity with the CYP2E1 activity was unexpected. Specifically, because no inhibition of vandetanib-*N*-oxide formation was produced by DDTC, an inhibitor of CYP2E1 (Table 1), and also because human recombinant CYP2E1 did not generate vandetanib-*N*-oxide (Figure 5A). Therefore, all these findings strongly suggest that FMO (predominantly FMO3) is the enzyme responsible for oxidation of vandetanib to vandetanib-*N*-oxide in human hepatic microsomes.

### 2.4. Oxidation of Vandetanib by Human Recombinant CYP and FMO Enzymes

The use of recombinant CYP enzymes expressed in Supersomes™ in combination with their reductase, POR, was another approach to examine the activity of individual human CYP enzymes to oxidize vandetanib. Under the experimental conditions used, several tested CYPs were capable of oxidizing vandetanib to *N*-desmethylvandetanib, but not to vandetanib-*N*-oxide (Figure 5A). Of the CYPs tested, five human CYPs were active in catalyzing the vandetanib oxidation to *N*-desmethylvandetanib (Figure 5A). Of these, human CYP3A4 was the most prominent enzyme oxidizing vandetanib with efficiencies up to one order of magnitude higher than those of the other CYPs (i.e., CYP2D6, 1A1, 2C8 and 3A5) (Figure 5A). Cytochrome b_5_, the heme protein participating in several functions of the CYP-monooxygenase system [25,26,27,28,29,30,31,32,33,34,35,36,37], influenced *N*-desmethylvandetanib formation catalyzed by several CYPs. The strongest impact of cytochrome b_5_ on this reaction was seen with human CYP3A4. The potency of human CYP3A4 to oxidize vandetanib was 8-fold higher (*p* < 0.001) when additionally stimulated by cytochrome b_5_. Cytochrome b_5_ also increased the formation of *N*-desmethylvandetanib in reactions catalyzed by CYP3A5 (7-fold; *p* < 0.001) and CYP2C8 (1.6-fold; *p* < 0.01). In contrast, essentially no such effect was observed when testing human CYP1A1 and 2D6 (Figure 5A). These results indicate that not only the expression and activities of these CYPs alone, but also the expression levels of cytochrome b_5_ can influence the metabolism of vandetanib in humans.

Similarly, the potency of human recombinant FMO1, FMO3 and FMO5 enzymes to metabolize vandetanib was evaluated using Supersomes™. In contrast to recombinant CYPs, recombinant FMOs were not able to oxidize vandetanib to *N*-desmethylvandetanib, but generated vandetanib-*N*-oxide (Figure 5B). FMO1 was over 5 times more efficient to form vandetanib-*N*-oxide than FMO3 (Figure 5B). The FMO3 enzyme is expressed in human liver, whereas FMO1 is not expressed in this human organ [38,39]. Therefore, it can be speculated that the formation of vandetanib-*N*-oxide is catalyzed by FMO3 in human hepatic microsomes.

### 2.5. Contributions of Individual CYPs to Oxidation of Vandetanib to N-Desmethylvandetanib in Human Livers

Based on the results showing the velocities of vandetanib oxidation to *N*-desmethylvandetanib in experimental systems containing recombinant CYP enzymes in Supersomes™ (Figure 6) and the relative amounts of CYP enzymes expressed in human livers [40,41,42,43,44], the contributions of individual CYPs to this reaction in human livers were evaluated. For these calculations we also took the presence and absence of cytochrome b_5_ into account. The highest contribution to *N*-desmethylvandetanib formation in human liver was attributed to CYP3A4 (~97.7%), followed by CYP2D6 (~1.7%) and CYP1A1, 2C8 and 3A5 (all less than ~0.3%) (Figure 6A). However, the pattern of CYP contributions in human liver clearly changed when the effect of cytochrome b_5_ on vandetanib oxidation was considered. In the presence of cytochrome b_5_, almost all *N*-desmethylvandetanib formation was attributed to CYP3A4 (~99.6%) with almost neglectable contributions (~0.4% in total) by other human CYPs (CYP2D6, 3A5, 1A1 and 2C8) (Figure 6B). Therefore, the impact of these CYPs on various responses to vandetanib treatment on individual patients should be negligible.

### 2.6. Binding Of Vandetanib to the Active Site of Compound I of Human CYP1A1, CYP2D6 and CYP3A4 

*N*-Demethylation of vandetanib precedes via the CYP-mediated attack on the carbon atom of the methyl group by oxygen of the Compound I of CYPs, which leads to the formation of the α-C-hydroxylation product that consequently decomposes to formaldehyde and *N*-desmethylvandetanib (Figure 1). Therefore, the binding affinity and orientation of the *N*-methyl group of vandetanib in the binary complex of vandetanib with the CYP active site, which is a prerequisite process for vandetanib-*N*-demethylation, should dictate the efficiency of individual CYPs to catalyze this reaction. Of the tested human CYP enzymes, CYP3A4, 2D6 and 1A1 were most effective in vandetanib *N*-demethylation (Figure 5A). Therefore, the binding affinity and orientation of the *N*-methyl group of vandetanib in the binary complex of vandetanib with the active sites of these CYPs were investigated. The flexible docking simulation found the probable productive binding position of vandetanib molecule in the active sites of human CYP1A1 and CYP2D6 (Figure 7A,B).

The binding positions in these CYPs (3.89 and 3.38 Å) are not optimal but facilitate *N*-demethylation, and consistently with our experiments the vandetanib is predicted to be better substrate for CYP2D6 than for CYP1A1 as it shows higher binding affinity and a more favorable reaction distance than CYP1A1 (Table 4). The active site of CYP3A4 possesses a large and flexible binding cavity, therefore we evaluated vandetanib interaction toward two structures of human CYP3A4, one crystallized without ligand presence and second co-crystallized with large ritonavir-like ligand. Using molecular docking techniques, we predicted several possible binding poses of vandetanib in the active site of human CYP3A4, and some of them were suitable for vandetanib-*N*-demethylation. The obtained productive binding poses were partially different (Figure 7C_1_,C_2_). This was not surprising because the binding sites of both structures were slightly different and substantially larger than volume of a single vandetanib molecule. The binding poses found were also showing less than optimal reaction distances of 3.12 and 3.45 Å, which is not consistent with our experimental findings that CYP3A4 demethylates vandetanib more efficiently than CYP1A1 and CYP2D6.

In order to explain the high efficiency of CYP3A4 in *N*-demethylation of vandetanib, we investigated the possibility of binding the second vandetanib molecule. The second vandetanib molecule was docked into the active site of CYP3A4 already containing one vandetanib molecule. For that we did not use the productive complex shown in Figure 7C_2_, but the energetically most favorable complex found during initial vandetanib docking, this ligand pose is rendered pink in Figure 7C_3_. The second ligand fitted well into the remaining space of the binding cavity accommodating a position near the heme residue. The productive position shows slightly lower binding energy than the first vandetanib molecule, but it shows minimal reaction distance for *N*-demethylation (Table 4, Figure 7C_3_).

Based on this theoretical approach it can be suggested that CYP3A4 in the complex with two vandetanib molecules bound to the active center is more efficient in catalyzing the *N*-demethylation of vandetanib than the classical binary complex of CYP3A4 with one vandetanib molecule. If this hypothesis is valid and the CYP3A4 enzyme binds more than one vandetanib molecule, this should lead to sigmoidal kinetics of vandetanib-*N*-demethylation. To further examine this hypothesis, we investigated the kinetics of the CYP3A4-mediated vandetanib oxidation. As shown in Figure 8 and Table 5, CYP3A4 oxidized vandetanib to *N*-desmethylvandetanib by a reaction exhibiting sigmoidal kinetics which supported our hypothesis originating from the molecular docking results. In contrast, CYP1A1 and 2D6 *N*-demethylate vandetanib by reactions exhibiting hyperbolic kinetics (Figure 8A,B), which is consistent with the models found where one molecule of vandetanib was bound to the active sites of CYP1A1 and 2D6.

The presence of cytochrome b_5_ in the CYP3A4 system significantly increased the rate of vandetanib oxidation but did not influence the type of kinetics (Figure 8C,D). Similar values of Hill coefficient of the reaction were found both for the free CYP3A4 form and CYP3A4 in complex with cytochrome b_5_ indicating that kinetics did not change. Cytochrome b_5_ strongly facilitated the velocity of the oxidative reaction and increased also the value of K_0.5_, indicating that a higher vandetanib concentration is required to attain half-maximal velocity. The K_0.5_ doubles with and without cytochrome b_5_, which suggests that possibly the allosteric effects of this heme protein may be affecting catalysis by altering the enzymatic affinity. The complex mechanism of the CYP3A4 kinetics for vandetanib oxidation to *N*-desmethylvandetanib should be considered. The CYP3A4 activity can be affected (i) by binding of two molecules of vandetanib to the active site, resulting in the sigmoid kinetics (positive cooperativity), and (ii) by binding of a non-substrate ligand acting as activator (cytochrome b_5_) at a site other than the substrate binding site (an allosteric modulation).

## 3. Discussion

The results of this study demonstrate that *N*-desmethylvandetanib and vandetanib-*N*-oxide, the two major oxidation products of vandetanib, are formed by human, rat, rabbit and mouse hepatic microsomes in vitro. However, the profile of vandetanib metabolites formed by microsomes isolated from different species varied significantly, especially in the amounts of generated vandetanib-*N*-oxide. Human hepatic microsomes were less efficient in the generation of vandetanib metabolites compared to hepatic microsomes isolated from rat, rabbit and mice. These findings can result from distinctive expression levels of CYPs and FMOs and/or from different efficiencies of orthologue forms of these enzymes in microsomes of these species that modulate the oxidation of vandetanib. The expression levels of the selective CYP and FMO enzymes were indeed significantly different in hepatic microsomes of human and the tested animal models. In human liver, the CYP3A4 enzyme is predominantly expressed and the major CYP of this organ (~30% of the CYP liver complement). This is in contrast to CYPs of the 2C subfamily (~15% for CYP2C9) and CYP1A2 (~13%) that are also highly expressed in this organ but are less abundant [40], whereas in rat liver, CYPs of the 2C subfamily are mainly expressed (~55% of the CYP rat liver complement) instead of enzymes of the CYP3A (~15%) and CYP1A (~2%) subfamilies. Likewise, in the livers of rabbits and mice, CYP2C are the major CYPs expressed in this organ [40,41,42,43,45,46]. Of the FMO enzymes, FMO3 (as well as FMO5) is known to be highly expressed in human and mouse liver [38,39], whereas FMO1 is not expressed in the human liver but it is the major FMO form expressed in the livers of rats and rabbits. Therefore, this distinct expression of individual liver CYP and FMO enzymes support the hypothesis on their impact on differences in vandetanib oxidation found in human and animal hepatic microsomes. Moreover, it should be emphasized that although the metabolic pathways of vandetanib were the same in human and several animal species [10,19], from the point of view of the CYP- and FMO-generated amounts of individual vandetanib metabolites, rat, rabbit and mouse experimental models differed significantly. Therefore, they are not suitable to mimic the metabolic situation in human.

In our study, we utilized several approaches to identify individual CYP and FMO enzymes oxidizing vandetanib. First, we investigated the effect of agents capable of decreasing the activities of individual CYPs and FMO. Ketoconazole, an inhibitor of CYP3A4, was the strongest inhibitor of the oxidation of vandetanib to *N*-desmethylvandetanib, while sulfaphenazole and quinidine that inhibit CYP2C and 2D, respectively, were less efficient. Inhibitors of other CYPs such as α-NF, diamantane and DDTC inhibiting CYP1A, 2B and 2E1, respectively, showed no inhibitory effects. Further, all tested CYP inhibitors were not capable of decreasing the formation of vandetanib *N*-oxide. Instead, the formation of this metabolite was strongly attenuated by methimazole, an inhibitor of FMO [30,31]. However, it is important to point out that results found with inhibitors are sometimes difficult to interpret because inhibitors can act more efficiently with one enzyme substrate than another. This might explain why itraconazol—which is considered a strong inhibitor of CYP3A4—exhibited a low potency to inhibit vandetanib oxidation to *N*-demethylation product, as described previously [10]. We therefore employed additional experimental approaches: (i) correlation analysis between the CYP- and FMO-catalytic activities in each microsomal sample with the amounts of vandetanib metabolite(s) formed by the same microsomes; and (ii) analysis of the oxidation of vandetanib to *N*-desmethylvandetanib and vandetanib-*N*-oxide by human recombinant CYPs and FMOs. Our results demonstrated a predominant role of CYP3A4 and FMO3 in the formation of *N*-desmethylvandetanib and vandetanib-*N*-oxide in human liver microsomes, respectively.

The fact that CYP3A4 is not only responsible for the majority of vandetanib oxidation to *N*-desmethylvandetanib but also the most abundant CYP found in human liver (when average 30% of total CYP in the human liver is considered) underlines the importance of this CYP in metabolizing vandetanib in human liver. However, the catalytic activity of CYP3A4 shows large interindividual variability depending on many factors. Expression differs from 30% to 60% of total CYP in human tissues but also genetic polymorphisms and modulation of CYP3A4 expression by other xenobiotics including drugs and the diet can contribute to interindividual differences in the CYP3A4-mediated pharmacokinetic profiles of vandetanib. Moreover, oxidation of vandetanib by CYP3A4 as the most prominent enzyme forming *N*-desmethylvandetanib is strongly stimulated by cytochrome b_5_. In addition to these phenomena, the mechanism of catalysis of vandetanib oxidation by CYP3A4 determines its high efficacy in this reaction. Molecular modeling indicated that binding of more than one molecule of vandetanib into the CYP3A4-active center occurs, which is responsible for the sigmoid kinetics of vandetanib oxidation and the high efficiency of CYP3A4 to *N*-demethylate vandetanib. Our finding that CYP3A4 activity can be modulated by binding of two molecules of the vandetanib substrate and by allosteric modulation by cytochrome b_5_ might be of great importance because the distribution of vandetanib in the human body can lead to differences in its concentrations in various human tissues. The sigmoidal kinetics of vandetanib binding to CYP3A4 strongly dictates its oxidation (*N*-demethylation) and might thus result in different response in patients to treatment with this drug. All these features can lead to different anticancer activity and/or adverse effects after vandetanib treatment. Indeed, this conclusion seems to be valid because distinct responses to vandetanib have been found in several patients suffering from medullary thyroid cancer treated with this drug [47]. However, in order to further support this conclusion, it will be important to monitor the marker enzyme activity of CYP3A4 in patients with the medullary thyroid cancer that have been shown to differ in their response to vandetanib and to correlate the phenotypic activity of CYP3A4 to clinical data that shows the patient’s responses to vandetanib treatment. In addition, future studies should consider drug–drug interactions related to CYP3A4 that are known to modulate the pharmacokinetic profiles of various anticancer drugs [21] and included when evaluating different patient’s responses to vandetanib treatment. These considerations, although beyond the scope of the present study, should help to optimize and personalize chemotherapy with vandetanib.

Even though vandetanib-*N*-oxide was determined to be a minor metabolite formed in human liver, our finding that FMO1 and FMO3 can efficiently catalyze the formation of this metabolite is of high significance. Namely, the formation of vandetanib-*N*-oxide significantly decrease the pharmacological effects of vandetanib, since it is more than 50-times less pharmacological active than vandetanib or *N*-desmethylvandetanib [10]. In contrast to human liver, the FMO1 enzyme that is most efficient to catalyze oxidation of vandetanib to vandetanib-*N*-oxide, is highly expressed in human kidney, lung, bone marrow, and gastrointestinal tract, where it can decrease the oxidation of vandetanib to this less efficient vandetanib metabolite. Therefore, the expression and enzymatic activities of FMO1 in these organs and FMO3 in human liver can determine the anticancer properties of vandetanib. Moreover, levels of FMOs and their activities have been reported to be modulated by developmental and hormonal changes rather than by exogenous compounds in several mammals including humans, dictating their final activities [48,49,50]. Since the FMO inhibitor methimazol is commonly used as a drug to treat hyperthyroidism in humans [31,51] it could be envisaged to be used as a modulator of FMO-mediated vandetanib oxidation thereby providing a new strategy to improve the anticancer efficiency of treatment of thyroid cancer patients with vandetanib. This remains a challenge for future research.

## 4. Materials and Methods

### 4.1. Chemicals and Material

Vandetanib was from LC Laboratories (Woburn, MA, USA). Ketoconazole, NADPH, methimazol and other chemicals were all purchased from Sigma-Aldrich (St. Louis, MO, USA). The purity of all chemicals met the standards of American Chemical Society, unless noted otherwise. All animal experiments were conducted in accordance with the Regulations for the Care and Use of Laboratory Animals (311/1997, Ministry of Agriculture, Czech Republic), which follows the Declaration of Helsinki. Male Wistar rats (~125–150 g; AnLab, Czech Republic), male C57BL/6 mice (5–8 g; CXR Bioscience Ltd., Dundee, UK), and male New Zealand rabbits (2.5–3 kg; AnLab, Czech Republic) were employed as animal models. Animals were placed in cages in temperature- and humidity-controlled rooms, acclimatized for five days and maintained at 22 °C with a 12 h light/dark period. Standardized diet and water were provided ad libitum. Rats, mice and rabbits used to prepare microsomal fraction from untreated animals were sacrificed by cervical dislocation, livers snap-frozen in liquid nitrogen and stored at −80 °C until further processing (see below). For controls, livers of the animals were removed immediately after sacrifice, divided into small pieces (~1 g), snap-frozen in liquid nitrogen, and stored at −80 °C until isolation of microsomal fractions. Pooled microsomes were used for all in vitro experiments in the present study and prepared from 10 livers/group for all animal species following established protocols [52,53]. Microsomal fractions were stored at −80 °C until analysis. Protein concentrations in the microsomal fractions were assessed using the bicinchoninic acid protein assay with bovine serum albumin as a standard [54]. Male human hepatic microsomes (pooled sample) (sample LOT: 3043885) were from Gentest Corp. (Woburn, MA, USA). Microsomes from livers of twelve human donors were obtained from Gentest Corp. (Woburn, MA, USA) (cat. no. HG43–1, HG103, HG74, HG93, HG24, HG27, HG23, HG32, HK31, HK34, HG03, and HG42). Each human microsomal sample has been characterized for CYP, FMO and protein contents and specific CYP activities by Gentest Corp. We reanalyzed each microsomal preparation for specific CYP, NADPH:CYP reductase (POR) and FMO activities by assays described in the protocols of Gentest Corp. Our data were similar to those reported by Gentest Corp. in their specification sheets. The content of CYP, specific CYP and FMO activities in each human hepatic microsomes [phenacetin-*O*-deethylation (CYP1A2), coumarin-7-hydroxylation (CYP2A6), (S)-mephenytoin-*N*-demethylase (CYP2B6), paclitaxel-6α-hydroxylation (CYP2C8), diclofenac-4’-hydroxylation (CYP2C9), (S)-mephenytoin-4′-hydroxylation (CYP2C19), bufuralol-1’-hydroxylation (CYP2D6), chlorzoxazone-6-hydroxylation (CYP2E1), testosterone-6β-hydroxylation (CYP3A4), lauric acid-12-hydroxylation (CYP4A), methyl-*p*-tolyl sulfide oxidase (FMO)] are shown in Table 2. Human recombinant enzymes were used in the forms of Supersomes™ and obtained from Gentest Corp. In Supersomes™ microsomal fractions were isolated from insect cells that are transfected with baculovirus constructs containing cDNA of human CYP enzymes (CYP1A1/2, 1B1, 2A6, 2B6, 2C8/9/18/19/, 2D6, 2E1, 3A4). These Supersomes™ also express NADPH:CYP oxidoreductase (POR). However, because they are microsomes (particles of broken endoplasmic reticulum), other enzymes (proteins) of the endoplasmic reticulum membrane (like NADH:cytochrome b_5_ reductase and cytochrome b_5_) are also expressed at basal levels in these Supersomes™. We also utilized Supersomes™ which over-expressed cytochrome b_5_, in a molar ratio of CYP to cytochrome b_5_ of 1 to 5. In Supersomes™ where cytochrome b_5_ was not over-expressed (see above) pure cytochrome b_5_ was added to reach a molar ratio of CYP to cytochrome b_5_ of 1 to 5. Reconstitution of Supersomes™ with purified cytochrome b_5_ was performed as described previously [35,55,56,57]. Human recombinant FMOs were also used in the forms of Supersomes™ (Gentest Corp.). These microsomal fractions were isolated from insect cells that are transfected with baculovirus constructs containing cDNA of human FMO1, FMO3 and FMO5 enzymes.

### 4.2. Oxidation of Vandetanib by Hepatic Microsomes and CYP Enzymes

Unless stated otherwise, incubation mixtures used to study vandetanib metabolism contained the following in a final volume of 500 μl for incubations containing hepatic microsomes and 250 μL for incubations with Supersomes™: 100 mM potassium phosphate buffer (pH 7.4), 1 mM NADPH, human, rat, rabbit or mouse hepatic microsomes (0.25 mg protein) (500 μl), or human recombinant CYPs or FMOs in Supersomes^TM^ (100 pmol) (250 μL) and 50 µM vandetanib dissolved in 5 µL dimethyl sulfoxide (DMSO). The reaction was initiated by adding vandetanib. In control incubations, either microsomes or CYP (FMO) or NADPH or vandetanib were omitted. Incubations were performed at 37 °C for 20 min in open plastic Eppendorf tubes; vandetanib oxidation was linear up to 30 min of incubation. After the incubations, 5 µL of 1 mM phenacetine (dissolved in methanol) was added as internal standard. The reaction was stopped by extraction with dichlormethane (2 × 1 mL). Extracts were evaporated, dissolved in 25 µl methanol and high-performance liquid chromatography (HPLC) analysis was used to separate vandetanib and its metabolites. HPLC conditions were as follows: Nucleosil® EC 100-5 C18 reverse phase column (150 × 4.6 mm, Macherey Nagel); the eluent was 0,5% *v*/*v* triethylamin in water (pH 3) containing 30% acetonitrile with a flow rate of 0.6 ml/min, detection was at 254 nm. Vandetanib metabolites separated HPLC were characterized by mass spectrometry (see further details below). Up to two vandetanib metabolites were detected with the retention times (r.t.) of 7.6 and 8.9 min, corresponding to *N*-desmethylvandetanib and vandetanib-*N*-oxide, respectively. The recoveries of vandetanib metabolites were approximately 95%.

### 4.3. Identification of Vandetanib Metabolites by Mass Spectrometry

Vandetanib metabolites were identified by direct infusion of diluted sample in 50% acetonitrile in water plus 1% acetic acid into the 12T solariX XR FT-ICR mass spectrometer (Bruker Daltonics, Bremen, Germany). The mass spectrometer was operating in positive ion mode. Calibration of the instrument was performed using 1% solution of sodium trifluoracetic resulting in accuracy below 2 ppm. Data acquisition and data processing were performed by ftmsControl 2.1.0 and DataAnalysis 4.2 (Bruker Daltonics).

### 4.4. Inhibition Studies

Inhibition studies in human liver microsomes were essentially conducted as reported previously [58]. The inhibitors employed were as follows: α-Naphthoflavone (α-NF), which inhibits CYP1A1 and 1A2; diamantane, which inhibits CYP2B; sulfaphenazole, which inhibits CYP2C; quinidine, which inhibits CYP2D; diethyldithiocarbamate (DDTC), which inhibits CYP2E1 and CYP2A; and ketoconazole, which inhibits CYP3A. The values of IC_50_ using 0.05–500 µM concentrations of inhibitors were also determined by the procedure described previously [59]. Inhibitors were dissolved in 5 µL DMSO, except for quinidine and α-naphthoflavone which were dissolved in methanol and except of DDTC, which was dissolved in water, yielding a final concentration of 50 µM in the incubation mixtures [equimolar concentrations of individual inhibitors with those of vandetanib (50 µM) were in incubation mixtures]. Inhibitors were incubated at 37 °C for 10 min with the 50 µM vandetanib prior to addition of NADPH, and then mixtures were incubated for further 20 min at 37 °C. The formation of vandetanib metabolites was analyzed by HPLC as described above.

### 4.5. Contributions of CYP Enzymes to N-Demethylation of Vandetanib to N-Desmethylvandetanibin in Human Livers

In order to calculate the contributions of individual CYPs to vandetanib oxidation (i.e. formation of *N*-desmethylvandetanib) in human livers, we utilized the velocities of vandetanib oxidation to *N*-desmethylvandetanib by the Supersomal CYP enzyme systems in combination with reported data on the expression levels of CYPs in human livers [40,41,42,43]. The contributions of these enzymes were calculated by relative activity factor because the activities of CYPs in Supersomes™ also need to be considered in addition to the relative contents of CYPs in the livers. Therefore, the contribution of each CYP enzyme that oxidizes vandetanib in livers was calculated by dividing the relative activity of each CYP oxidizing vandetanib [r.a._cypi_] (rate of vandetanib oxidation to *N*-desmethylvandetanib) multiplied by the amounts of this CYP in tissues examined, by the total relative activities (∑[r.a._cypi_]) of all CYPs oxidizing this substrate. Of CYPs in human liver, CYP3A4 is the major enzyme present in this human organ (~30% of the CYP hepatic complement), followed by CYP2C9 and 1A2 (~15% and ~13%, respectively), while CYP2C19, 2E1 2A6, 2D6, 2C8 and 3A5 are present in human liver at levels that range from ~2.5% to ~8.5% (see Reference [40] for an overview).

### 4.6. Molecular Docking of Vandetanib into Compounds I of Human CYP1A1, 2D6 and 3A4

The X-ray based coordinates of human CYP1A1 (PDB ID 4I8V) [60], CYP2D6 (PDB ID 3TDA) [61], CYP3A4 ligand-free form (2.8 Å resolution, PDB ID 1W0E) [62] and inhibitor (ritonavir-like ligand) bound form (2.42 Å resolution, PDB ID 6BD7) [63] were used as starting structures for modeling of vandetanib interactions with the activated state—Compound I of selected human CYPs. During structure preparation, hydrogen atoms were added and crystallographic water and ligand molecules were removed, usual protonation states and Gasteiger partial charges were assigned to all residues, except for the atomic charge of the ferric ion of the heme cofactor, for which a value more consistent with a metal in octahedral coordination was used [64]. The geometries and charges of a vandetanib ligand in neutral form were predicted using ab initio calculations on the Hartree–Fock level of theory in conjunction with the 6–31+G(d) basis set. All ab initio calculations were carried out with program Gaussian 09 [65]. Autodock v4.2.6 was employed for flexible docking using procedure described previously [66]. All rotatable bonds of the residues S122, F123, F224, F258, D313, N222, L496, D320, T321 (in CYP1A1) or F120, F483, L213, E216 (in 2D6) and vandetanib ligand were allowed to rotate freely. We carried out an extensive search (1000 docking runs per system) of the most preferred binding modes of vandetanib molecule within a 60×60×60 Å grid-box centered on the substrate binding cavity. Resulting structures showing RMSD lower than 2.0 Å were grouped and finally sorted by binding free energy of the best binding structure within each cluster. A set of binding modes with similar binding energies was found for every system as a result. We assume that only the orientations with a sufficiently short distance between carbon of the *N*-methoxy group of vandetanib and the activated oxygen atom in the CYP Compound I would facilitate the *N*-demethylation reaction.

### 4.7. Statistical Analysis

Data are expressed as mean ± SD. Data was analyzed using GraphPad Prism 7 using ANOVA with post-hoc Tukey HSD Test. *p*-value < 0.05 was considered as significant. Statistical associations between CYP- and FMO-linked catalytic activities in human hepatic microsomal samples and amounts of *N-*desmethylvandetanib and vandetanib-*N*-oxide formed by the same microsomes were determined by linear regression using Statistical Analysis System software version 6.12. Correlation coefficients (*r*) were based on a sample size of twelve for human microsomes. All *p* values are two-tailed and considered significant at the 0.01 level.

## 5. Conclusions

*N*-desmethylvandetanib and vandetanib-*N*-oxide were identified as the two major oxidation products of vandetanib, the drug used for the treatment of symptomatic or progressive medullary thyroid cancer due to the RET tyrosine kinase mutation occurring in this type of thyroid cancer. They are generated by human, rat, rabbit and mouse hepatic microsomes in vitro. In the human hepatic microsomal systems, CYP3A4 >>>> CYP2D6 >> CYP1A1 are important in the formation of *N*-desmethylvandetanib, while FMO3 oxidizes vandetanib to vandetanib-*N*-oxide. The oxidation of vandetanib by CYP3A4, the most prominent enzyme catalyzing formation of *N*-desmethylvandetanib, is stimulated by cytochrome b_5_. We further elucidated the molecular mechanism of vandetanib-*N*-demethylation catalyzed by CYP3A4, which might be of great importance to optimize the treatment regimen of the drug. Molecular modeling (in silico docking) and sigmoid kinetics of the reaction demonstrated that two molecules of vandetanib are bound to the CYP3A4-active center. This process results in a shift of orientation of the *N*-methyl group of vandetanib to be located closer to oxygen of the CYP3A4 Compound I leading to an optimal reaction distance for *N*-demethylation. The novel results showing the allosterically modulated CYP3A4-catalyzed oxidation of vandetanib, in addition to a new finding of a decrease in FMO-mediated generation of vandetanib-*N*-oxide by methimazol, are challenges for further investigation and potential utilization to improve treatment of thyroid cancer with vandetanib.

## Figures and Tables

**Figure 1 ijms-20-03392-f001:**
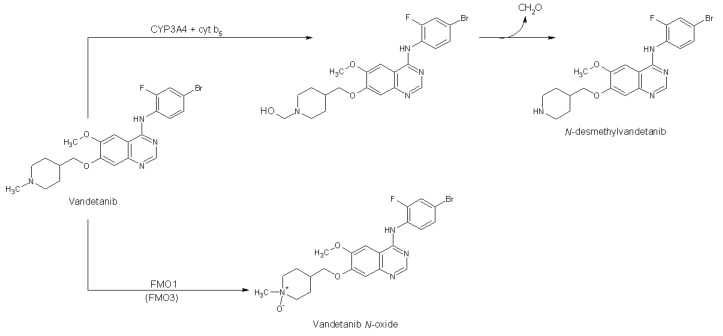
Vandetanib oxidation by CYP and FMO enzymes. Cytochrome b_5_ (cyt b_5_).

**Figure 2 ijms-20-03392-f002:**
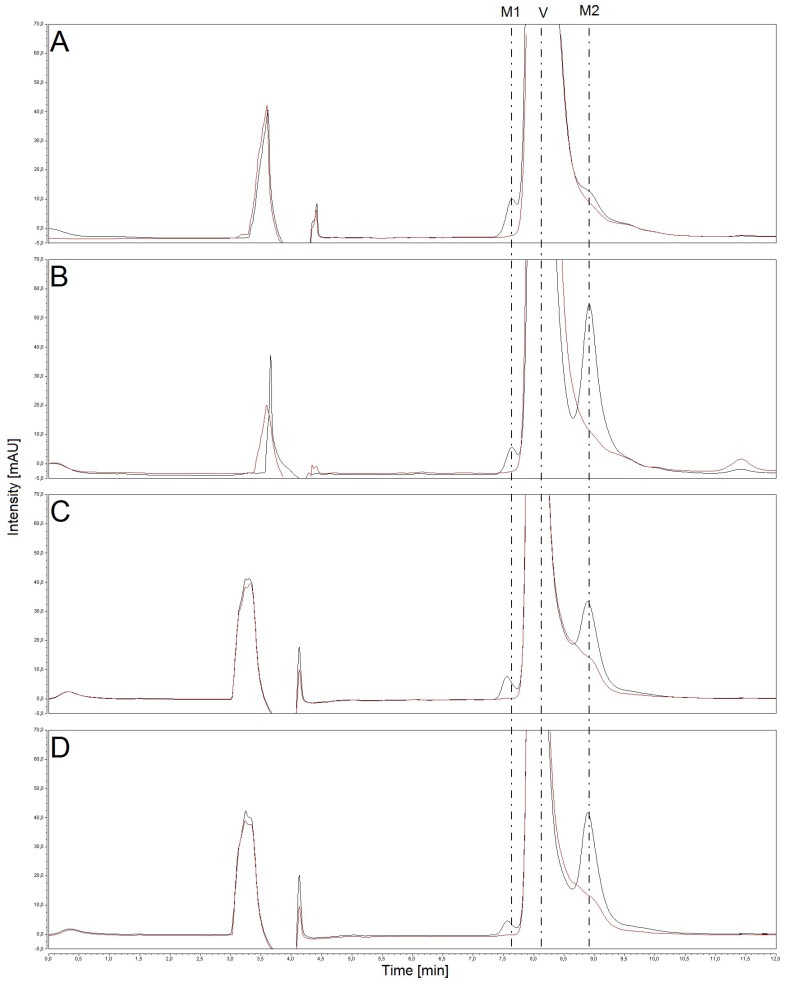
Separation of vandetanib and its metabolites formed by human (**A**), rat (**B**), rabbit (**C**) and mouse (**D**) hepatic microsomes in the presence of NADPH (black line) and without NADPH (control) (red line) using HPLC analysis. M1—*N*-desmethylvandetanib; V—vandetanib; M2—vandetanib-*N*-oxide.

**Figure 3 ijms-20-03392-f003:**
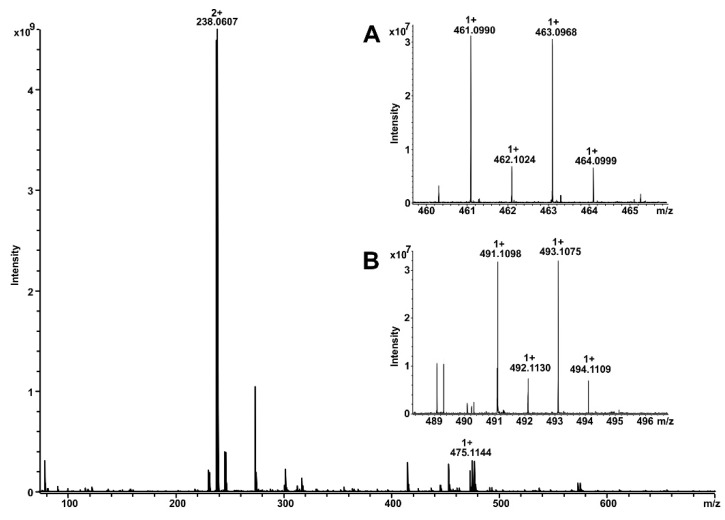
MS spectrum of vandetanib products measured by 12T electrospray ionization coupled with Fourier-transform ion cyclotron resonance (ESI-FTICR). Vandetanib was observed as singly and doubly charged ion at *m*/*z* 475.1144 (1+) and m/z 238.0607 (2+) (Δppm 1.0). Insert **A** represents the detail mass spectrometry (MS) spectrum of the vandetanib metabolite *N*-desmethylvandetanib at *m*/*z* 461.0990 (Δppm 1.5). Insert **B** represents the detail MS spectrum of the vandetanib metabolite *N*-oxide of vandetanib at *m*/*z* 491.1098 (Δppm 1.8).

**Figure 4 ijms-20-03392-f004:**
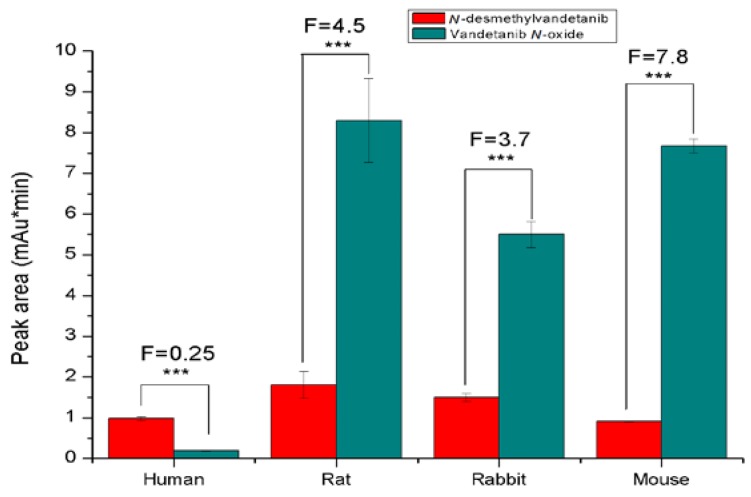
Oxidation of vandetanib by human, rat, rabbit and mouse hepatic microsomes. Values represent mean ± SD of three independent in vitro incubations (*n* = 3). F, fold difference between formation of *N*-desmethylvandetanib and vandetanib-*N*-oxide. **** p* < 0.001 significant differences between formation of *N*-desmethylvandetanib and vandetanib-*N*-oxide (ANOVA with post-hoc Tukey honestly significant difference (HSD) Test).

**Figure 5 ijms-20-03392-f005:**
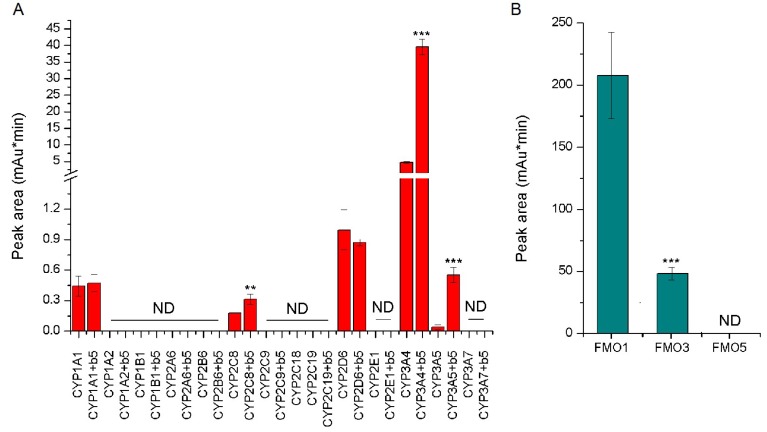
Oxidation of vandetanib to *N*-desmethylvandetanib by human recombinant CYPs (**A**) and FMOs (**B**). Values represent mean ± SD of three independent in vitro incubations (*n* = 3). *** *p* < 0.001, ** *p* < 0.01 significant differences between formation of *N*-desmethylvandetanib by CYP enzymes with and without cytochrome b_5_ (b5) in Panel (**A**) and *** *p* < 0.001 significant differences between formation of vandetanib-*N*-oxide in Panel (**B**). (ANOVA with post-hoc Tukey HSD Test). ND—not detected.

**Figure 6 ijms-20-03392-f006:**
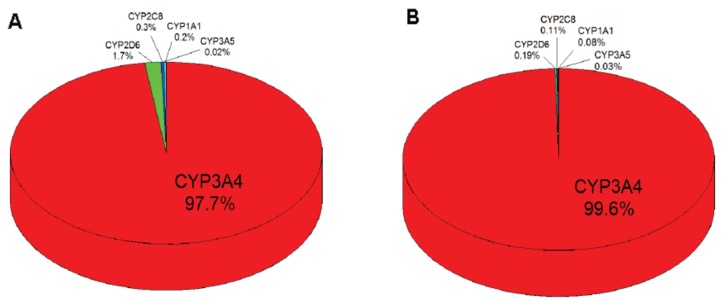
Contributions of human hepatic CYP enzymes to formation of *N*-desmethylvandetanib without consideration of the effect of cytochrome b_5_ (**A**) and with that of cytochrome b_5_ (**B**).

**Figure 7 ijms-20-03392-f007:**
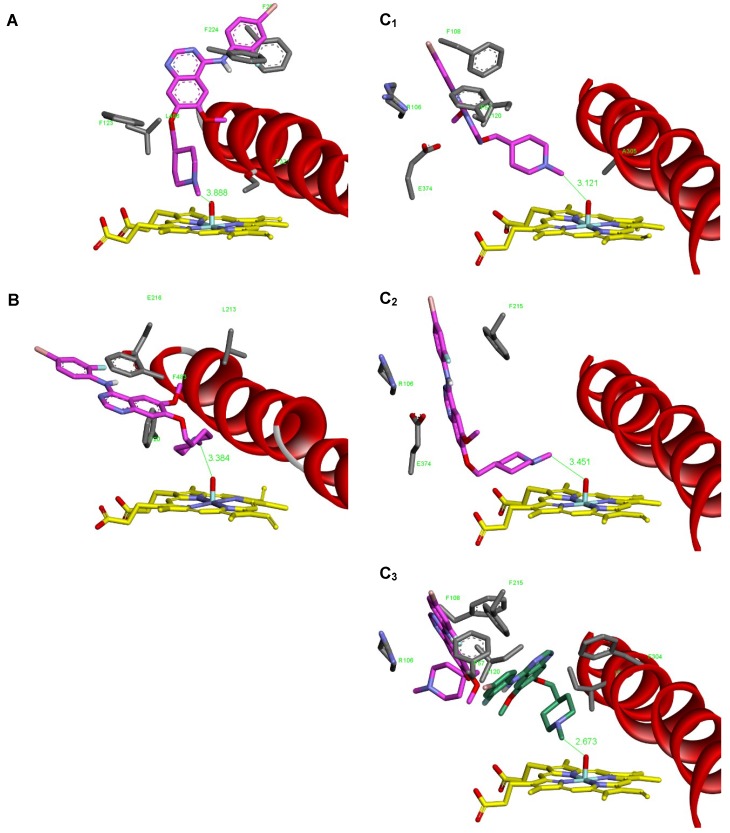
The binding orientations found in molecular docking calculations facilitating oxidative demethylation of *N*-methyl group of vandetanib in human CYP1A1, CYP2D6 and CYP3A4 represented by X-ray structures CYP1A1 4I8V (**A**), CYP2D6 3tda (**B**), CYP3A4 1W0E (**C_1_**); CYP3A4 6BD7 (**C_2_**); and CYP3A4 6BD7 containing second vandetanib molecule (green) (**C_3_**). Vandetanib molecule (pink/green) and heme (yellow) residues are rendered as bold sticks. Side-chains of amino acid residues closely interacting with vandetanib are shown in gray. Red ribbon represents a part of helix I.

**Figure 8 ijms-20-03392-f008:**
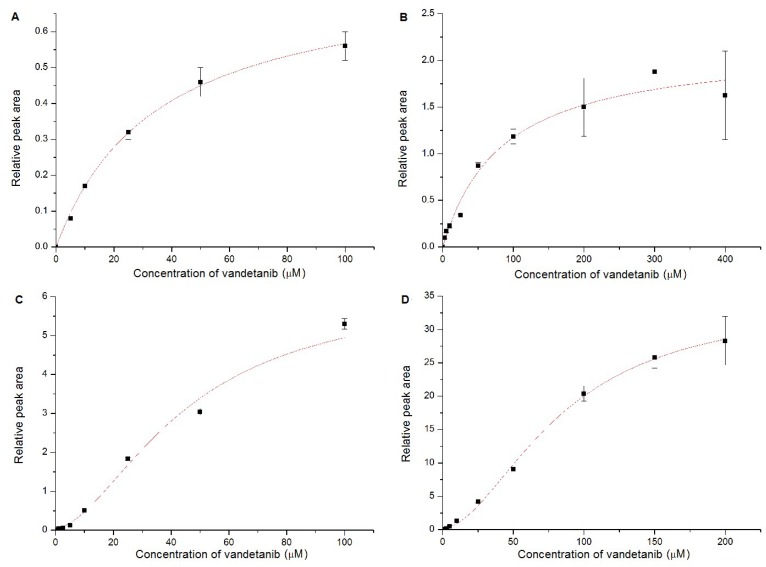
Enzyme kinetics of vandetanib oxidation to *N*-desmethylvandetanib catalyzed by CYP1A1 (**A**), 2D6 (**B**), 3A4 (**C**) and this enzyme in the presence of cytochrome b_5_ (**D**). Values represent means ± SD of three independent in vitro incubations (*n* = 3).

**Table 1 ijms-20-03392-t001:** Effects of CYP and FMO inhibitors on oxidation of vandetanib by human liver microsomes.

Enzymes	Inhibitor ^a^	% of Inhibition	IC_50_ (μM)
		*N*-Desmethyvandetanib Formation	
**CYP**	**α-Napththoflavone (CYP1A)**	0	NA ^b^
	Diamantane (CYP2B)	0	NA
	Sulfaphenazole (CYP2C)	38 ± 4 ***	NA
	Quinidine (CYP2D)	20 ± 4 **	NA
	DDTC (CYP2A, CYP2E1)	0	NA
	Ketoconazole (CYP3A)	98 ± 3 ***	2 ± 0.2
		**Vandetanib-*N*-oxide Formation**	
**FMO**	Methimazol (FMOs)	79 ± 4 ***	6 ± 0.5

^a^ CYPs for which compounds acts as their specific inhibitors are listed in brackets. Equimolar concentrations of individual inhibitors and vandetanib (50 μM), 0.1 nmol of microsomal CYP for CYP inhibition and 0.25 mg of microsomal protein for FMO inhibition were in incubation mixtures. Values represent mean ± SD of three independent in vitro incubations (*n* = 3). ^b^ NA, not applicable. *** *p* < 0.001, ** *p* < 0.01, statistically different from data of controls, without inhibitors (ANOVA with post-hoc Tukey HSD Test).

**Table 2 ijms-20-03392-t002:** CYP- and FMO-dependent catalytic activities and amounts of *N*-desmethylvandetanib and vandetanib-*N*-oxide in human liver microsomes.

^a^ No.	^b^ Total CYPs	^c^ POR activity	^d^ Cyt b_5_	^e^ CYP1A2	^e^ CYP2A6	^e^ CYP2B6	^e^ CYP2C8	^e^ CYP2C9	^e^ CYP2C19	^e^ CYP2D6	^e^ CYP2E1	^e^ CYP3A4	^e^ CYP4A11	^e^ FMO	^f^ M1	^f^ M2
**HG03**	290	450	380	170	2000	51	200	1700	44	110	1800	6100	1600	^g^ NM	3.639	^h^ ND
**HG103**	340	210	790	310	440	7.2	39	2300	23	65	1100	2200	1600	1400	1.021	0.004
**HG24**	260	260	550	1700	1500	35	190	3000	41	NM	2300	4000	1800	1500	1.698	0.004
**HG32**	170	330	580	730	520	0.68	20	450	4.8	46	1200	2000	680	920	0.870	ND
**HG42**	670	510	500	700	2200	150	480	1600	7,4	95	1600	15000	1400	2000	8.887	0.015
**HG43**	270	210	640	580	770	14	25	1800	440	4	780	4600	1800	920	1.830	ND
**HG74**	220	200	600	520	360	13	130	2100	55	120	1400	2700	1300	1200	0.714	0.004
**HG93**	430	320	450	691	350	43	270	2200	75	49	2800	2800	1800	3500	0.966	0.112
**HK23**	380	380	700	960	1100	24	160	2100	110	140	2100	6800	780	2500	4.137	0.008
**HK27**	300	450	730	1320	1320	31	180	480	460	130	3000	4910	1110	2230	1.762	ND
**HK31**	580	540	770	1220	2160	8.1	130	1690	172	3,4	1660	8210	2010	3020	4.124	0.015
**HK34**	500	460	890	1000	1500	39	220	1900	45	100	6000	5200	1100	2700	3.193	ND

^a^ Human hepatic microsomal samples; ^b^ Total CYP in pmol CYP/mg protein; ^c^ POR activity in cytochrome c reductase (pmol/mg protein), ^d^ Cyt b_5_ was determined spectrophotometrically. ^e^ CYP- and FMO-specific activity in pmol product/(mg protein × min); ^e^ Phenacetin-*O*-deethylase—CYP1A2; ^e^ Coumarine-7-hydroxylase—CYP2A6; ^e^ (S)-Mephenytoin-*N*-demethylase—CYP2B6; ^e^ Paclitaxel-6α-hydroxylase—CYP2C8; ^e^ Diclofenac-4′-hydroxylase—CYP2C9; ^e^ (S)-Mephenytoin-4′-hydroxylase—CYP2C19; ^e^ Bufuralol-1′-hydroxylase—CYP2D6; ^e^ Chlorzoxazone-6-hydroxylase—CYP2E1; ^e^ Testosterone-6β-hydroxylase—CYP3A4-; ^e^ Lauric acid-12-hydroxylase—CYP4A11; ^e^ Methyl-*p*-tolyl sulfide oxidase—FMO; ^f^
*N*-desmethylvandetanib (M1) and vandetanib-*N*-oxide (M2) in peak area/(mg protein × min); the detection limit for the formation of metabolite M2 was < 0.002 (in peak area/(mg protein x min); ^g^ Not measured; ^h^ ND—not detectable (levels below the limit of detection).

**Table 3 ijms-20-03392-t003:** Correlation coefficients (*r*) among CYP- and FMO-linked catalytic activity and amounts of *N*-demethylvandetanib and vandetanib-*N*-oxide formed in microsomes.

	Total CYPs	POR	Cyt b5	CYP1A2	CYP2A6	CYP2B6	CYP2C8	CYP2C9	CYP2C19	CYP2D6	CYP2E1	CYP3A4	CYP4A11	FMO
**^a^ M1**	0.786 **	0.700	−0.114	0.018	0.778 **	0.826 ***	0.747 **	−0.034	−0.168	0.165	0.043	0.984 ***	0.002	0.270
**^a^ M2**	0.166	0.035	−0.5743	−0.157	−0.339	0.093	0.288	−0.039	0.088	−0.343	0.714 **	−0.200	0.315	0.736 **

*** *p* < 0.001; ** *p* < 0.01. ^a^ M1—*N*-desmethylvandetanib; ^a^ M2—vandetanib-*N*-oxide

**Table 4 ijms-20-03392-t004:** The predicted binding free energies and distances facilitating *N*-demethylation of vandetanib bound in selected CYPs.

Simulated System	The Most Stable Productive Orientations of the Neutral Form of Vandetanib in the Complex with CYPs	
	Estimated Free Energy of Binding (kcal/mol)	O(Comp I)-*N*-CH_3_Distance [Å]^a^
CYP1A1 (4I8V)	−8.64	3.89
CYP2D6 (3TDA)	−10.21	3.38
CYP3A4 (1W0E)	−9.96	3.12
CYP3A4 (6BD7)	−9.63	3.45
CYP3A4 (6BD7) + second vandetanib molecule	−9.30	2.67

^a^ Distance between the carbon in the *N*-methyl group of vandetanib and oxygen atom on heme iron in the complex of an activated CYP enzymes (Compound I), see Figure 7.

**Table 5 ijms-20-03392-t005:** The characteristics of the kinetics of the oxidation of vandetanib to *N*-desmethylvandetanib by human CYP1A1, 2D6 and 3A4.

CYP Enzyme	Kinetical Characteristics		
	^a^ V_Max_	^b^ K_0.5_	^c^ Hill Coefficient
CYP1A1	0.76 ± 0.02	35.12 ± 2.75	NA
CYP2D6	2.16 ± 0.15	84.36 ± 18.14	NA
CYP3A4	6.17 ± 1.32	45.02 ± 13.62	1.73 ± 0.56
CYP3A4+cyt b_5_	34.32 ± 1.59	83.65 ± 5.55	1.83 ± 0.13

Values represent means ± SD of three independent in vitro incubations (*n* = 3). ^a^ V_Max_—maximal velocity of *N*-desmethylvandetanib formation (peak area/min/nmol CYP); ^b^ K_0.5_—the substrate concentration corresponding to half-maximal velocity (µM); ^c^ Hill coefficient–Hill cooperativity index. Hill cooperativity index of *n* = 1.73 for CYP3A4 and 1.83 for CYP3A4 in the presence of cytochrome b_5_ (cyt b_5_) show that two substrate molecules can be bound to the active site of the CYP3A4 enzyme. NA—not applicable.

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
