# Peer review of "Identification of Human Enzymes Oxidizing the Anti-Thyroid-Cancer Drug Vandetanib and Explanation of the High Efficiency of Cytochrome P450 3A4 in its Oxidation"

_ijms, 2019, doi:10.3390/ijms20143392_

Round 1
Reviewer 1 Report
Indra et al. describe reaction phenotyping of vandetanib in microsomes and with recombinant enzymes. This reaction has been well studied, but the authors describe a novel mechanism of 3A4 activation (b5) and cooperativity of two-substrate binding. I have several comments that I hope may improve the precision, clarity and presentation of the manuscript.
I found the result with CYP3A5 to be interesting. Were the HLMs genotyped for 3A5? Did the authors consider using Cyp3cide instead of ketoconazole to elucidate contributions in HLMs between the two 3A enzymes? Allelic distributions are widely different in 3A5 and may provide additional insight into differential PK found in patients.
In Table 2 the authors refer to M2 as not detected in several samples. It would be helpful to determine the limit of detection (LOD) of your assay and report that values are below the LOD.
Figure 8/Table 5: On Line 362 of the text, the authors remark that b5 did not change the binding affinities. However the Km doubles with and without b5. This suggests that possibly the allosteric effects of b5 may be effecting catalysis by altering the enzymatic affinity. Consider proposing an explanation for this.
Figure 3. Please at the Δppm measurement to the reported masses. This will provide mass accuracy. Also I would change "product" to metabolite in lines 143 & 144
Author Response
Indra et al. describe reaction phenotyping of vandetanib in microsomes and with recombinant enzymes. This reaction has been well studied, but the authors describe a novel mechanism of 3A4 activation (b5) and cooperativity of two-substrate binding.
Response: We thank the reviewer for this comment.
I have several comments that I hope may improve the precision, clarity and presentation of the manuscript.
I found the result with CYP3A5 to be interesting. Were the HLMs genotyped for 3A5? Did the authors consider using Cyp3cide instead of ketoconazole to elucidate contributions in HLMs between the two 3A enzymes? Allelic distributions are widely different in 3A5 and may provide additional insight into differential PK found in patients.
Response: We thank the reviewer for the comment. Indeed, allelic distributions are widely different in CYP3A5 that is considered to provide additional insight into differential PK found in patients. The HLMs used in the experiments were not genotyped for CYP3A5; no liver samples from the human donors of liver microsomes were available. Although human recombinant CYP3A5 can catalyze oxidation of vandetanib to N-desmethylvandetanib, the efficiency of this enzyme in this reaction is very low; more than two- and almost three-orders of magnitude lower than CYP3A4, with and without the presence of cytochrome b5 (see Fig. 5). We calculated the contributions of CYP3A5 in the formation of N-desmethylvandetanib from these efficiency data and the average absolute amounts of CYP3A5 in human liver (human liver microsomes) found previously (Rendic and DiCarlo, Drug Metab. Rev. 29, 413, 1997; Westlind-Johnsson et al., Clin. Pharmacol. Ther. 79, 339, 2003; Liu et al., Proteomics, 14, 1943, 2014) and shown to be almost four orders of magnitude lower than those of CYP3A4. Using this approach, the contribution of CYP3A5 to the formation of N-desmethylvandetanib was found to be only ~0.03 and ~0.02 % of the CYP liver complement with and without cytochrome b5, respectively (Fig. 6). In contrast, 97.7 and 99.6 % of the CYP liver complement were attributed to the CYP3A4-mediated reaction without and with cytochrome b5, respectively (Fig. 6). We took these data into account and did not analyze the effect of Cyp3cide on vandetanib oxidation to N-desmethylvandetanib by HLMs. Nevertheless, utilizing Cyp3cide is an interesting idea we suppose to utilize in our future studies investigating the contributions of CYP3A4 and 3A5 to the oxidation of other CYP3A substrates, which are oxidized by CYP3A5 more efficiently.
In Table 2 the authors refer to M2 as not detected in several samples. It would be helpful to determine the limit of detection (LOD) of your assay and report that values are below the LOD.
Response: The detection limit (LOD) for formation of vandetanib N-oxide (metabolite M2) in our assay was included into the legend to Figure 2. The text that values are below the limit of detection was included into the revised version of the manuscript (see page 7, lines 213-214).
Figure 8/Table 5: On Line 362 of the text, the authors remark that b5 did not change the binding affinities. However the Km doubles with and without b5. This suggests that possibly the allosteric effects of b5 may be effecting catalysis by altering the enzymatic affinity. Consider proposing an explanation for this.
Response: We thank the reviewer for this comment. Indeed, the KM doubles with and without cytochrome b5, which suggests that possibly the allosteric effects of this heme protein may be affecting catalysis by altering enzymatic affinity. The complex mechanism of the CYP3A4 kinetics for vandetanib oxidation to N-desmethylvandetanib should be considered. CYP3A4 activity can be affected (i) by binding two molecules of vandetanib to the active site resulting in the sigmoid kinetics (positive cooperativity), and (ii) by binding non-substrate ligands acting as activator (e.g. cytochrome b5) at a site other than the substrate binding site (an allosteric modulation). Therefore, combinations of the allosteric and cooperative effects might be responsible for results found in the reaction leading to the formation of N-desmethylvandetanib by CYP3A4. This explanation is now included in the revised version of the manuscript (see page 12, lines 368-376). The KM was substituted by K0.5 (half-maximal velocity) in Table 5, better suited to sigmoidal kinetics.
Figure 3. Please at the Δppm measurement to the reported masses. This will provide mass accuracy. Also I would change "product" to metabolite in lines 143 & 144.
Response: The values of Δppm were included into the legend of Figure 3. Further, as suggested by the reviewer the word “product” was changed to “metabolite”.
This reviewer suggested that the moderate English changes are required. Therefore, the corrections were carried out by the English speaking expert.

Reviewer 2 Report
authors have studied the metabolism of vanditanib in microsomes as well as recombinant protein preparations and used invitro assays and molecular modeling to find a mechanism for binding of this drug to CYp3A4. Study is original and well conducted. It is acceptable after minor changes.
Some mention in the introduction section about the role of cytochrome b5 in CYP3A4 reaction and its contribution to activity is needed.
In the microsomal preparations, amount of b5 and POR activity as a cytochrome c reduction assay should be included to give some idea about exact ratios / amounts of these proteins in the preparations.
The figure 7 needs revision. Please white background and label the residues, drug, heme and nearby amino acids to give some idea of binding pocket.
Author Response
Authors have studied the metabolism of vandetanib in microsomes as well as recombinant protein preparations and used in vitro assays and molecular modeling to find a mechanism for binding of this drug to CYP3A4. Study is original and well conducted. It is acceptable after minor changes.
Response: We thank the reviewer for the positive feedback.
Some mention in the introduction section about the role of cytochrome b5 in CYP3A4 reaction and its contribution to activity is needed.
Response: We agree with the reviewer. The role of cytochrome b5 in CYP3A4 reaction and its contribution to CYP activity was included into the Introduction section of the manuscript (see page 3, lines 119-122).
In the microsomal preparations, amount of b5 and POR activity as a cytochrome c reduction assay should be included to give some idea about exact ratios / amounts of these proteins in the preparations.
Response: The amounts of cytochrome b5 and POR activity, measured as a cytochrome c reduction, were included into Table 2.
The figure 7 needs revision. Please white background and label the residues, drug, heme and nearby amino acids to give some idea of binding pocket.
Response: The Figure 7 was revised as suggested by the reviewer.
